# Influence of the Processing Method on the Properties of Ti-23 at.% Mo Alloy

**Patrycja Sochacka** [ID]**, Andrzej Miklaszewski** [ID]**, Kamil Kowalski** [ID] **and Mieczyslaw Jurczyk \*** [ID]

Institute of Materials Science and Engineering, Poznan University of Technology, Jana Pawla II 24, 61-138 Poznan, Poland

**\*** Correspondence: mieczyslaw.jurczyk@put.poznan.pl; Tel.: +48-61-665-3508

**Abstract:** In this paper, binary β type Ti-23 at.% Mo alloys were obtained by arc melting as well as by mechanical alloying and powder metallurgical process with cold powder compaction and sintering or, interchangeably, hot pressing. The influence of the synthesis method on the microstructure and properties of bulk alloys were studied. The produced materials were characterized by an X-ray diffraction technique, scanning electron microscopy and chemical composition determination. Young's modulus was evaluated with nanoindentation testing method based on the Oliver and Pharr approach. The mechanically alloyed Ti-23 at.% Mo powders, after inductively hot-pressed at 800 °C for 5 min, allowed the formation of single Ti(β) phase alloy. In this case, Young's modulus and Vickers hardness were 127 GPa and 454 $HV_{0.3}$, respectively. Among the examined materials, the porous (55%) single-phase scaffold showed the lowest indentation modulus (69.5 GPa). Analytical approach performed in this work focuses also on the surface properties. The estimation includes the corrosion resistance analyzed in the potentiodynamic test, and also some wettability properties as a contact angle, and surface free energy values measured in glycerol and diiodomethane testing fluids. Additionally, surface modification of processed material by micro-arc oxidation and electrophoretic deposition on the chosen samples was investigated. Proposed procedures led to the formation of apatite and fluorapatite layers, which influence both the corrosion resistance and surface wetting properties in comparison to unmodified samples. The realized research shows that a single-phase ultrafine-grained Ti-23 at.% Mo alloy for medical implant applications can be synthesized at a temperature lower than the transition point by the application of hot pressing of mechanically alloyed powders. The material processing, that includes starting powder preparation, bulk alloy transformation, and additional surface treatment functionalization, affect final properties by the obtained phase composition and internal structure.

**Keywords:** material processing; mechanical alloying; titanium β alloys; phase transformation; powder metallurgy; X-ray diffraction; fluoroapatite coating; corrosion resistance; contact angle measurements

## 1. Introduction

Titanium and the Ti-6Al-4V alloy remain the main metallic biomaterials for orthopaedic and dental applications [1–4]. Young's modulus of these biomaterials is, however, much higher than that of the human bone (20–27 GPa) [5]. In order to reduce the undesirable (SSE) stress shielding effect and the mismatch of Young's modulus, some metallic elements such as Zr, Nb, Mo, Ta have been proposed and added to titanium for new Ti(β) or near Ti(β) alloys, such as Ti5Al13Ta [6], Ti5Al5Mo5V3Cr [7,8], Ti5Al5Mo5V3Cr, Ti5Al5Mo5V3Cr1Zr [9], Ti14Zr16Nb [10], and Ti23Zr25Nb [10].

Recent reports have shown that Ti–Mo alloys have great potential for surgical applications [5,11,12]. The studies and evaluation of the phase transformations and mechanical properties of Ti–Mo alloys

have also concluded that the phase composition and mechanical properties remain different for these biomaterials with a changeable Mo content [5,12,13].

The phase diagram of Ti–Mo shows the molybdenum solubility limits in the titanium matrix drag by a temperature reliance [14]. The research confirms that the higher addition of Mo forms a stable Ti(β) phase in Ti-based alloys, and eventually also increases hardness and decreases the elasticity modulus [10]. The microcrystalline Ti–Mo alloys with molybdenum content from 3.2 to 12 at.% were synthesized by the arc melting method [13]. The 3.2 and 8 at.% additions allows only α and β phases to form and characterize with a low elasticity modulus; however, for 4.5, 6, and 7 at.% contents the same research report a high E modulus owing to some presence of the ω phase in the alloy structure.

Independently, the properties of Ti–Mo alloys (with 4–19 wt.%) synthesized by the laser alloying method were studied [15]. Two-phase biomaterials were obtained for the Mo content in a range 4 to 8 wt.% (martensite and Ti(β) phase). For a concentration higher ≥ 10 wt.%, the obtained alloys were a single β phase.

It has been indicated that the composition and heat treatment strongly influence the microstructure and mechanical properties of Ti–Mo alloys [8,9,16]. Additionally, a high cooling rate enables the production of Ti(β) phase materials for the concentration of 10% of Mo. A strong influence of the structure on the elasticity modulus and hardness were also confirmed for samples aged at 450 °C, following the α and ω phase composition.

It is well known that the properties of Ti-based alloys can be enhanced not only by a changeable composition but also by their microstructure modification [17–27]. For nearly a decade, the application of nano- or ultrafine-grained materials have become very popular in implantology [21,25,27]. The enhancement of properties of Ti-based biomaterials can be obtained by a microstructure control. For example, the top-down approach method can be used: severe plastic deformation (SPD) or mechanical alloying (MA) [21,27].

The MA technique allows the improvement of the material properties by the obtainment of the nanocrystalline or ultrafine structure. As the example, the hardness increase based on the mechanism of grain boundary strengthening [24] can be distinguished. The MA process by cold welding and fragmentation of powder materials leads finally to grain refinement. The additionally obtained nanoscale also creates an inherent morphological change and this, as reports confirm, may influence adhesion, proliferation, and growth cells activity [28].

Recently, Ti–xx at.% Mo (xx = 10–35) alloys have been prepared by mechanical alloying and the powder metallurgy approach [18,29]. The Mo addition to titanium and proper heat treatment of nearly amorphous powders allows the synthesizing of a Ti(β) alloys. In this work, the arc-melting, as well as mechanical alloying and powder metallurgical process based on cold powder compaction and sintering or, alternatively, hot pressing (HP), was applied for the obtainment of the Ti(β)–type (Ti-23 at.% Mo) alloy. For this study, this alloy was labelled Ti23Mo. Additionally, for the Ti23Mo alloy, the micro-arc oxidation (MAO) and electrophoretic deposition (EPD) approaches were applied and led to the formation of apatite and fluorapatite (FA) layers, which improved analyzed surface properties compared to the base sample. The crystal structure, microstructure, composition, porosity, corrosion resistance, mechanical, and surface wetting properties of the bulk synthesized alloy were studied. To date, no attention has been paid to the influence of the processing method on the evolution of the properties in the Ti23Mo biomaterial.

## 2. Materials and Methods

The present work concludes the research results carried out on the Ti-23 at.% Mo alloy synthesized by different methods. For clearance, obtained materials were marked as follows:

-AM—arc melted;

-AMA800—arc melted and annealed 800 °C/24 h;

-HP—hot-pressed at 800 °C/5 min;

-CP—cold-pressed and sintered at 800 °C/0.5 h;

　　　-CP + NH$_4$HCO$_3$—cold-pressed with NH$_4$HCO$_3$ and sintered in a vacuum of $10^{-2}$ Pa in two steps: (i) Space-holder particles burn out at 175 °C for 2 h, (ii) heat-treatment at 1150 °C for 10 h [23];

　　　-CP + MAO—cold-pressed and sintered at 800 °C/0.5 h, samples next treated by micro-arc oxidation;

　　　-CP + MAO + EPD—cold-pressed and sintered at 800 °C/0.5 h, samples next treated by micro-arc oxidation and electrophoretic deposition.

## 2.1. Sample Preparation

　　　Powders of titanium (<45 µm, 99.9%, Alfa Aesar, Karlsruhe, Germany) and molybdenum (44 µm, 99.6%, Sigma Aldrich, Karlsruhe, Germany) were used as primary materials. Binary β type Ti23Mo alloys were synthesized by arc melting as well as mechanical alloying and the powder metallurgical process with a CP and HP approach.

　　　In the first approach, the microcrystalline Ti23Mo ingot was obtained by arc-melting of the powders on a water-cooled copper pot under Ar. The powders of Ti and Mo were weighed, mixed and placed into the die (8 mm in diameter), uniaxially pressed (600 MPa) and finally arc-melted. The obtained alloy was re-melted three times for homogeneity. Additionally, the arc melted alloy was annealed at 800 °C for 24 h.

　　　In the second approach, the ultra-fine grained materials were synthesized by mechanical alloying and the powder metallurgical process. The MA was performed under Ar (99.999% putity) by the application of the SPEX 8000 Mixer Mill (SPEX SamplePrep, Metuchen, NJ, USA). The total milling time was 48 h. The Ti and Mo powders were weighed, blended, and insert into stainless steel vials in the glove box (LabMaster 130) filled with automatically controlled argon atmosphere (O$_2$ < 2 ppm and H$_2$O < 1 ppm).

　　　A ball to powder ratio was set to 10:1. The size of the powders after 48 h of MA was 13.5 nm according to the Williamson–Hall approach calculation method and it was the subject of the detailed investigation with powder processing and preparation in our earlier research [18]. So prepared precursor powders were next processed by the powder metallurgy (CP and HP approach). In the CP approach, precursors were inserted into a die and uniaxially pressed at a pressure of 600 MPa. For sintering, the green compacts were placed in argon-filled quartz tubes and heated for 1 h to 800 °C and then kept at temperature for 30 min. Obtained sinters dimensions were 8 mm diameter and 4 mm height. For the HP samples, an induction module was used for a conductive die heating by the Joule's heat generated on its surface. The HP was carried out at 800 °C for 300 s within a heating step of 800 s in the vacuum (50 Pa) with acting pressure of 60 MPa. A detailed description of the hot pressing procedure was included in the authors' previous work [18].

　　　Additionally, the mechanically alloyed Ti23Mo powders were mixed with ammonium hydrogen carbonate (AHC)-CH$_4$HCO$_3$ (500–800 µm, 98%, Alfa Aesar) used as the space-holder filler. The powder mixture prepared by the above-mentioned recipe was uniaxially pressed at the pressure of 400 MPa. Obtained samples dimensions were close to that one from the earlier procedure. The green compacts were next sintered in a vacuum of $10^{-2}$ Pa in two steps. Firstly, the space-holder particles were burned out at 175 °C for 2 h secondly, the compacts were heat-treated at 1150 °C for 10 h as was performed in the authors' previous research [23]. The porous (55%) Ti23Mo scaffold was obtained by the addition of 35 wt.% AHC to the powder mixture.

　　　Additionally, the surface treatment functionalization based on MAO and EPD process was performed. The oxidation process [30–32] was realized under Atlas Sollich potentiostat (300 V/3 A) equipment control, at a constant voltage of 250 V vs. open circuit potential for 3 min. As the electrolyte, an aqueous solution of 0.01 M Ca$_3$(PO$_4$)$_2$, 0.5 M citric acid was chosen.

　　　Fluorapatite particles were hydrothermally prepared by the recipe given in the [33]. Subsequently, the FA suspension in ethanol was magnetically stirred for 30 min followed by 15 min ultrasonic treatment. After the MAO process, electrophoretic deposition [34] of FA was accomplished at the negative voltage −200 V for 1 min in the fluoroapatite suspension in ethanol.

### 2.2. Materials Characterization

The crystallographic structure examination at the preparation and final processing stages was realized by the Panalytical Empyrean equipment with the copper anode 1.54Å (Almelo, The Netherlands). A detailed description of the structural analysis and evaluation methodology was included in the authors' previous work [17,23].

Additionally, for the lattice parameter estimation and phase quantitative analysis, the Rietveld approach was used. The applied estimation realized in the Maud software involved a simulation of the diffraction patterns based on the structural models for: Ti($\alpha$) and Ti($\alpha'$) (ref. code 01-071-4632), Ti($\beta$) (ref. code 01-074-7075).

A scanning electron microscopy (SEM, VEGA 5135 Tescan, Brno, Czech Republic) was used to characterize obtained samples microstructure; additionally, for non-etched surfaces observation, optical microscopy was used (Olympus GX51, Shinjuku, Tokio, Japan). For chemical composition determination, the energy dispersive spectrometer adapter (EDS, PTG Prison Avalon, Princeton Gamma Tech., Princeton, NY, USA) was used, calibrated with a typical Cu calibration procedure.

The density of the obtained sinters was determined by the Archimedes drainage method. For the sample porosity measurement, formula $P = (1 - \rho/\rho_{th}) \times 100\%$ was used, where $\rho$ is the density of the porous material and $\rho_{th}$ is its corresponding theoretical density calculated based on the rule of mixtures. For the hardness measurement of the bulk samples, Vickers microhardness testing approach was used (HV). The average value was calculated from the 10 separate indents on each sample for the load of 300 g during 10 s.

Indentation Hardness (HM) and modulus (EIT) of the non-etched Ti23Mo samples, was evaluated by a CSM Instruments nanoindenter with the Berkovich diamond tip [35]. The Depth-sensing indentation technique was used for the measurements of:

-indentation Martens Hardness (HM)

-indentation Modulus (EIT) based on the Oliver and Pharr [36] approach.

A detailed description of the measurements realized at the room temperature based on the ISO 14577 standard for $F = 0.3$ N per 20 s and $C = 5.0$ s parameters was included in the authors' previous work [37].

Additionally, the corrosion resistance properties of the obtained samples were evaluated. The surface of the sample was prepared by grinding in water up to 600 grit. Next, the samples were cleaned ultrasonically with ethanol for 15 min and dried in a cold air stream. For the measurements, the samples were first immersed in the Ringer's solution for 1 h vs. open circuit potential (OCP). The Ag/AgCl electrode was used as the reference electrode in the electrochemical cell. Three measurements for each sample were carried out. The weight loss by polarization was calculated using Faraday's law:

$$W = \frac{EW \cdot I_{corr} \cdot A}{F} \tag{1}$$

$W$—weight loss [g·s$^{-1}$];
$I_{corr}$—corrosion current [$\mu$A·cm$^{-2}$];
$EW$—equivalent weight [g·mol$^{-1}$];
$A$—surface [cm$^2$];
$F$—Faraday constant [A·s·mol$^{-1}$]

The samples were also immersed in the Ringer's solution for 14 days to measure the weight loss ($W$ by weight loss). Then, they were ultrasonically cleaned for 3 min in a solution of 30 vol.% $HNO_3$ + 3 vol.% HF (ASTM B 600-91). The weight of the sample was evaluated before and after the immersion (Kern ABT 120-5DM).

The MAO and EPD modified sample surfaces were investigated after drying and 24 h desiccator storage. The XRD analysis was carried out after a single treatment. The obtained diffractograms were

processed by the background subtraction and peaks position determination. Additionally, for the obtained surface development the SEM imaging technique was also used.

For the surfaces wetting investigations the samples which for the additional MAO and EPD processes were not pursued the grinding (up to 400 grit) and ultrasonically rinsing in acetone for cleaning and preparation were followed. The contact angle (CA) of the obtained surfaces was analysed by the optical system with a digital camera (Kruss-DSA25, KRÜSS GmbH, Hamburg, Germany) and estimated by added software (Kruss-Advanced 1.5, KRÜSS GmbH, Hamburg, Germany). A static surface contact angle measurements were carried out with glycerol (99.9%, Chemland, Poland) and diiodomethane (99.9%, Chemland, Poland) testing fluids. The CA values were determined from the geometrical shape of the droplets using the Young–Laplace function and manual baseline correction. Surface free energy (SFE) of analysed samples was estimated from the Owens, Wendt, Rabel, and Kaelble (OWRK) model used today most frequently. It is based on Fowkes and uses contact angles of two liquids with known polar and disperse component of SFE. A detailed description of the surfaces wetting analysis was included in the authors' previous work [20,38].

## 3. Results and Discussion

The aim of the current study was the synthesis of the Ti23Mo alloy with a beta type structure by arc-melting as well as mechanical alloying and the powder metallurgical process with a CP or HP approach and the evaluation of the properties as a function of microstructure. Additionally, material volumetric and surface functionalization changes were also investigated.

The crystal structure changes during mechanical alloying of Ti23Mo were studied earlier [29]. The typical (hkl) indexes of the titanium and molybdenum remain visible after 15 min of MA. After 5 h of milling, the new MoTi phase is formed. 15 h of milling allows the formation of a new Ti(β) phase. During processing, an energy transfer to a powdered material results in an increase of defects density with a subsequent subgrains formation, which may eventually even lead to material amorphization [39]. Processed for 48 h, the powder mixture evinces a strongly amorphous character with a crystallite size value estimated by the Williamson–Hall UDM approach close to 13 nm with an increases microstrain level at the range of $5.6 \times 10^{-3}$ [18]. The XRD analysis confirms for MA processed powder the phase transition possibility from (α) to (β) form during synthesis. What earlier research also shows is that the molybdenum content and the milling time remain crucial parameters responsible for transformation control [29].

The processed arc-melted, CP and HP sinters samples spectra were gathered in Figure 1. The XRD analysis that includes the microcrystalline arc melted (Figure 1a), arc melted and annealed (800 °C/24 h) (Figure 1b), hot-pressed and sintered (Figure 1c), cold-pressed and sintered (Figure 1d), as well as the scaffold samples with the porosity of 55% (Figure 1e), was revealed. The sintering results in the formation of bulk materials. A single-phase, β-type, Ti23Mo alloy and a Ti23Mo scaffold with the porosity of 54.7% were obtained by the HP approach and CP with the addition of ammonium hydrogen carbonate and sintering. The arc-melted sample is also a pure Ti(β) phase-type alloy. For the Ti-23Mo sample (arc-melted and annealed at 800 °C for 24 h), due to high saturation, another Ti(α')phase was detected in the β phase region (Figure 1b). Its content equals 17.0%. The obtained two-phase sample structure is characterized by a homogenous low porosity microstructure. On the other hand, the cold-pressed and sintered sample mostly remain a β-type one, with some (5.2%) content of the second Ti(α) phase. The structural parameters of the synthesized Ti23Mo alloys are summarized in Table 1. The porous and nearly fully light-reflective Ti23Mo alloy surfaces are shown in Figure 2 for which the EDS results have confirmed their chemical composition (Figure 3). The small content of an impurity (α-Fe) was detected in the MA sintered samples, due to the erosion of the milling media. The hot pressing method allows the synthesizing of a bulk Ti23Mo alloy of very low porosity (Table 2, Figure 2c). As can be seen, the porosity heavily depends on the processing method.

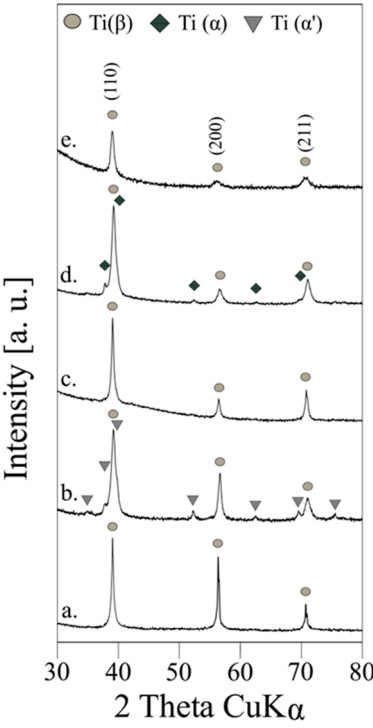

**Figure 1.** XRD spectra of Ti23Mo alloys obtained by different processing approaches: Arc melted (**a**), arc melted and annealed at 800 °C/24 h (**b**), MA for 48 h and: hot pressed at 800 °C/5 min (**c**), cold-pressed and sintered at 800 °C/0.5 h (**d**) and cold-pressed with $NH_4HCO_3$ and sintered at 1150 °C/10 h (**e**).

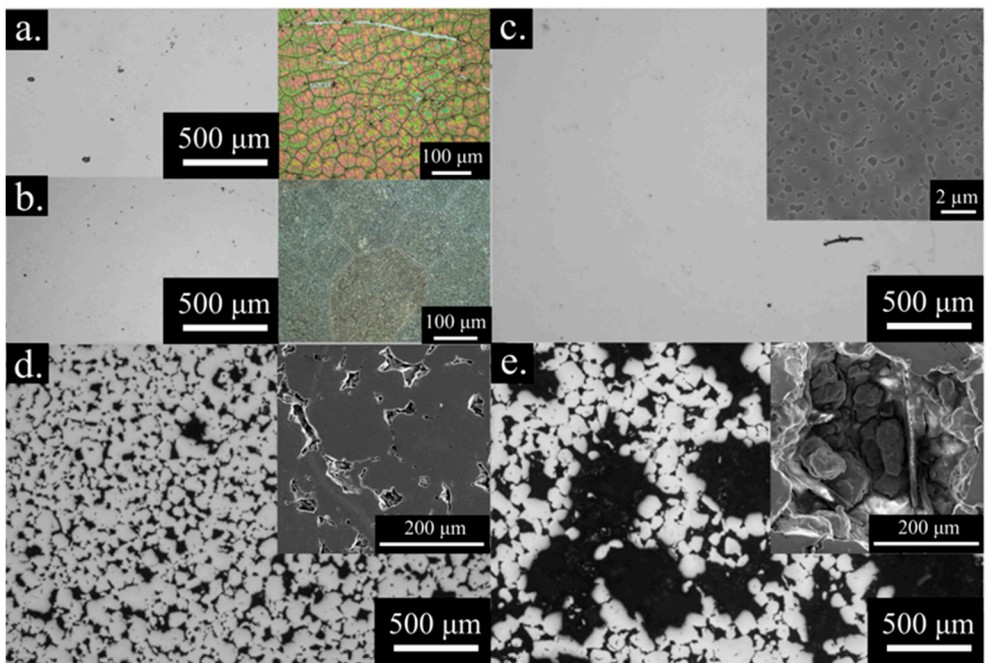

**Figure 2.** Optical and SEM microphotographs of Ti23Mo alloys obtained by different processing approaches: Arc melted (**a**), arc melted annealed at 800 °C/24 h (**b**), MA for 48 h and: hot pressed at 800 °C/5 min (**c**), cold-pressed and sintered at 800 °C/0.5 h (**d**) and cold-pressed with $NH_4HCO_3$ and sintered at 1150 °C/10 h (**e**).

**Table 1.** The structural parameters of Ti23Mo alloys synthesized by different processing approaches, estimation based on the Rietveld approach with phase amounts values (*A*%).

| Material | Sig | $R_{wp}$ (%) | $R_{exp}$ (%) | Phase Type | A (%) | Structural Parameters | | |
|---|---|---|---|---|---|---|---|---|
| | | | | | | *a* (Å) | *c* (Å) | *V* (Å³) |
| AM | 2.666797 | 10.484024 | 3.9313207 | Ti(β) | 100.0 | 3.2619(1) | - | 34.707(3) |
| AMA800 | 1.573098 | 6.3340626 | 4.0264897 | Ti(β) | 82.98 | 3.2475(1) | - | 34.250(3) |
| | | | | Ti(α′) | 17.02 | 2.9712(11) | 4.7592(18) | 36.385(41) |
| HP | 2.3671894 | 8.234684 | 3.4786754 | Ti(β) | 100.0 | 3.2570(0) | - | 34.550(1) |
| CP | 1.7763894 | 5.885216 | 3.3130217 | Ti(β) | 94.80 | 3.2453(0) | - | 34.178(1) |
| | | | | Ti(α) | 5.20 | 2.9700(10) | 4.7540(28) | 36.316(45) |
| CP + NH₄HCO₃ | 1.1901675 | 5.851319 | 4.9163833 | Ti(β) | 100.0 | 3.2615(2) | - | 34.659(7) |

**Table 2.** Theoretical density ($\rho_{th}$), calculated density ($\rho_{cal}$), and porosity (*P*) of Ti23Mo alloy obtained by different processing approach.

| Material | $\rho_{th}$ (g/cm³) | $\rho_{cal}$ (g/cm³) | *P* (%) |
|---|---|---|---|
| AM | 6.695 ± 0.164 | 6.674 ± 0.180 | 0.31 ± 0.06 |
| AMA800 | 6.695 ± 0.164 | 6.691 ± 0.167 | 0.06 ± 0.01 |
| HP | 6.700 ± 0.177 | 6.684 ± 0.195 | 0.24 ± 0.08 |
| CP | 6.688 ± 0.219 | 5.040 ± 0.671 | 24.64 ± 0.45 |
| CP + NH₄HCO₃ | 6.690 ± 0.095 | 3.046 ± 0.578 | 54.47 ± 0.67 |

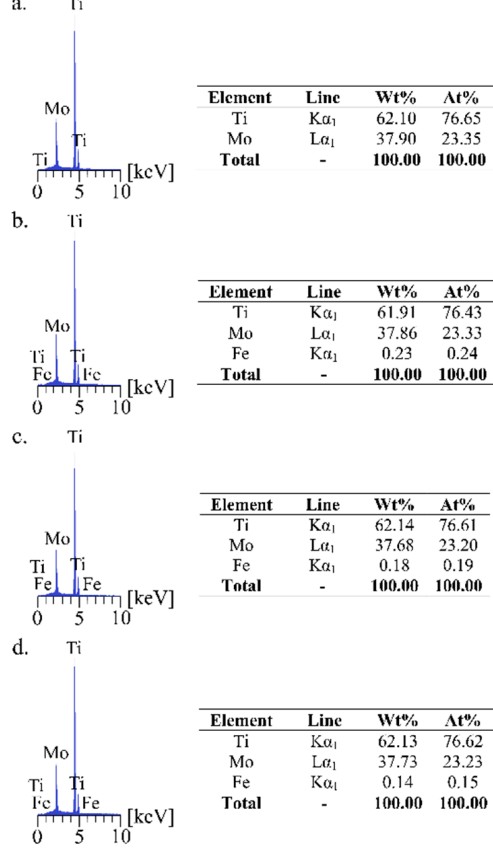

**Figure 3.** The amount of elements in Ti23Mo alloy: Arc melted (**a**), hot pressed (800 °C/5 min) (**b**), cold-pressed at 800 °C/0.5 h (**c**), cold-pressed with NH₄HCO₃ at 1150 °C/10 h (**d**) with their EDS spectra.

The SEM analysis was conducted to confirm the ultrafine-grained structure in the hot-pressed sinters. The highly magnified BSE mode microphotograph of the hot-pressed Ti23Mo sample presented in Figure 2c (top right-hand corner), confirmed the microstructure size range.

The Martens hardness (HM), Vickers microhardness ($HV_{0.3}$), and indentation modulus ($E_{IT}$) were shown for all the indentations selected (Table 3, Figure 4). The microhardness of the sintered samples shown a variety of distribution that was related to the microstructural changes. For example, the Vickers microhardness for the microcrystalline (arc-melted) and the cold-pressed and sintered (800 °C/0.5 h) alloys reached 547 and 366 $HV_{0.3}$, respectively. In the case of the hot-pressed Ti23Mo alloy, the Vickers microhardness increased to 454 $HV_{0.3}$ and it was almost three times higher compared to microcrystalline Ti (180 $HV_{0.3}$). A ten-times lesser force (300 mN), applied using the indentation depth-sensing technique, shows different results from those of the Vickers and Martens hardness measurements. A smaller examination area, resulting from the application of the Berkovich indenter, reflects more correctly the the material response by avoiding the porosity shear but suffers in the case of a multiphase material of higher scatter. The data shown in Table 3 for the above-mentioned relation, show very close results shared for higher data transparency, confirm sample homogeneousness, and the control of results for standard deviation. Alloying and reduction of structural objects following MA, strengthening of the solid solution as well as the grain refinement mechanism following sintering, allow an improvement of the analyzed material properties.

**Table 3.** Vickers hardness ($HV_{0.3}$), Martens hardness (HM), and Young's modulus ($E_{IT}$) of the Ti23Mo alloys obtained by different processing approaches.

| Material | $HV_{0.3} \pm \sigma$ | $HM \pm \sigma$ (N/mm$^2$) | $E_{IT} \pm \sigma$ (GPa) |
|---|---|---|---|
| AM | 547 ± 7 | 4289.4 ± 28.2 | 141.2 ± 2.6 |
| AMA800 | 366 ± 6 | 3270.7 ± 47.9 | 142.8 ± 4.3 |
| HP | 454 ± 6 | 3531.2 ± 32.7 | 127.3 ± 1.2 |
| CP | 366 ± 19 | 3093.1 ± 111.4 | 104.9 ± 10.5 |
| CP + NH$_4$HCO$_3$ | 397 ± 17 | 2880.7 ± 184.3 | 69.5 ± 8.9 |

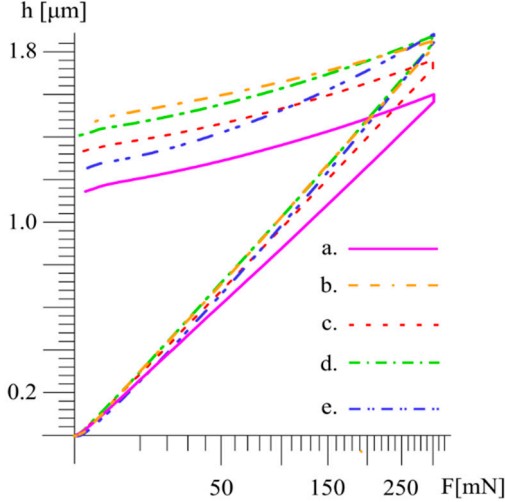

**Figure 4.** Depth-load (h-F) nanoindentation curves of Ti23Mo alloys obtained by different processing approaches: Arc melted (**a**), arc melted and annealed at 800 °C/24 h (**b**), MA for 48 h and hot pressed at 800 °C/5 min (**c**), cold-pressed and sintered at 800 °C/0.5 h (**d**) and cold-pressed with NH$_4$HCO$_3$ and sintered at 1150 °C/10 h (**e**).

The load-displacement curves of the synthesized Ti23Mo alloys and the scaffold are shown in Figure 4. In the case of the hot-pressed alloy (Table 3), the Young's modulus was 127.29 GPa, which was considerably lower than that of the Ti (140 GPa), Co–Cr–Mo alloy (210 GPa), and 316L stainless steel

(200 GPa) commonly used in orthopedic applications [3]. The obtained two-phase Ti23Mo structure in the case of the arc-melted and annealed (800 °C/24 h) alloy is characterized by the highest reported *E* modulus values (Figures 2 and 4). The lower average value of EIT characterizes the porous material of the porosity of 55% (69.5 GPa).

The mechanical properties of the Ti ($\alpha'$) phase can be controlled by a process of cold working and subsequent low-temperature heat treatment. For example, an $\alpha'$ martensite high strength alloy Ti-10Nb-2Mo-4Sn (wt.%) could be obtained [40].

The analyzed values of the modules for differently synthesized Ti23Mo alloys show a relation based on phase composition and shear of the microstructural elements. The first relation manifests with the modules drop, due to the appearance of a single beta phase or a diminishment of the second phase for the increased value of the stabilizing elements or different treatment approach. The second relation shows a direct connection to the microstructural features based on its internal structure. The material consistency analyzed in terms of porosity induced intentionally or being the effect of processing, through its volumetric amount influences the resultant value of the module. The above-mentioned relation corresponding with the obtained results indicates a possibility of module shaping.

The authors' earlier results have shown that the crystal structure of the solution treated alloys is sensitive to the Mo contents [18]. When it increases, the $\beta$ phase becomes the only dominant one. Molybdenum stabilizes the $\beta$-Ti structure and may suppress the $\omega$-phase transition, which, in many compositions, may exhibit unique properties such as the shape memory effect [41,42]. The presented results demonstrate that different synthesis methods of the $\beta$ Ti-based alloys may influence the phase composition as well as the final sinters properties.

It is noteworthy that the elasticity modulus of the studied alloy can be significantly reduced by the introduction of a porous structure [43]. The interconnected porous structure may facilitate the transportation of body fluid and the attachment of the implant to the surrounding bone tissue. For example, the bulk Ti23Mo scaffold with the porosity of 55% has a lower Young's modulus (69.5 GPa) compared to microcrystalline titanium (Table 3). This scaffold exhibited wide cavities of 250–500 μm in diameter (Figure 2e). The optimal pore size for the cell attachment, differentiation, and ingrowth osteoblasts and vascularization is approximately 200–500 μm [44]. In general, great variations in the elastic modulus and the plateau stress of the scaffolds can be achieved by different chemical compositions, pore morphologies, pore sizes and their distributions, shape and thickness of the struts, different compressive strength test parameters employed (sample geometry, size, loading speed) as well as by different fabrication methods [45].

It is well known that the elastic modulus of materials remains sensitive to the phase/crystal structure as well as inherent system confirmation. It has been demonstrated that metastable phases such as $\alpha'$, $\alpha''$, $\omega$ and $\beta$ can be formed during quenching from the high-temperature $\beta$ field, depending on the content of the $\beta$-stabilizers (e.g., Zr, Nb, Mo, Ta, etc.) [17,18,46,47].

Due to extremely small grain sizes, ultrafine-grained metals enhance physicochemical, mechanical and biological properties compared with the corresponding materials of a microcrystalline grain size [19,40,48]. A small degree of residual porosity after powder compaction also plays a role in the cell adhesion.

Earlier, Collings, and Gebel studied the elastic modulus in Ti–Mo alloys [49]. Firstly, due to solid solution strengthening, the *E* modulus increased lightly until 120 GPa at 7.5 wt.% Mo. A further increase in the Mo content caused a decrease of Young's modulus corresponding to the transition to the $\beta$ phase microstructure. A minimum value of 75 GPa was achieved for the Ti-13 wt.% Mo alloy. A further increase in the Mo content caused a slight increase in the elastic modulus to values up to 90 GPa. These results were in good agreement with those obtained on alloys prepared by casting [50].

The analyzed additional corrosion resistance behavior of the Ti–Mo samples (Table 4 and Figure 5) in the Ringer's solution, shows the relation resulting from the material's porosity and their chemical composition. The obtained corrosion resistance results show the best values for the hot-pressed, and arc-melted and annealed (800 °C/24 h) samples. The potential values, analyzed separately from the

corrosion curves, indicate a possible increase in the speed of the reactions that may take place on the surface, particularly the one with easier access of the liquid environment to the material substructure. The presence of porosity in the analyzed samples plays an essential role in the corrosion behavior of the material. Separate immersing test results confirmed that the weight loss analysis remains in close relation to the earlier discussion.

**Table 4.** Estimated from Tafel extrapolation corrosion potential ($E_{corr}$) and current ($I_{corr}$) with calculated weight loss from Faraday law (W by polarization) and after 14 days immersing in Ringer solution environment (W by weight loss) of Ti23Mo alloys obtained by different processing approaches.

| Material | $E_{corr}$ (V) | $I_{corr}$ ($\mu A \cdot cm^{-2}$) | W by Polarization ($\mu g \cdot day^{-1}$) | W by Weight Loss ($\mu g \cdot day^{-1}$) |
|---|---|---|---|---|
| AMA800 | −0.556(2) | 0.3913(66) | 4.7(1) | 3.5(5) |
| HP | −0.276(6) | 0.3333(355) | 4.0(4) | 1.7(2) |
| CP | −0.610(3) | 4.506(705) | 53.7(8) | 73.3(6) |
| CP + NH$_4$HCO$_3$ | −0.511(4) | 1.139(313) | 13.6(2) | 42.3(8) |

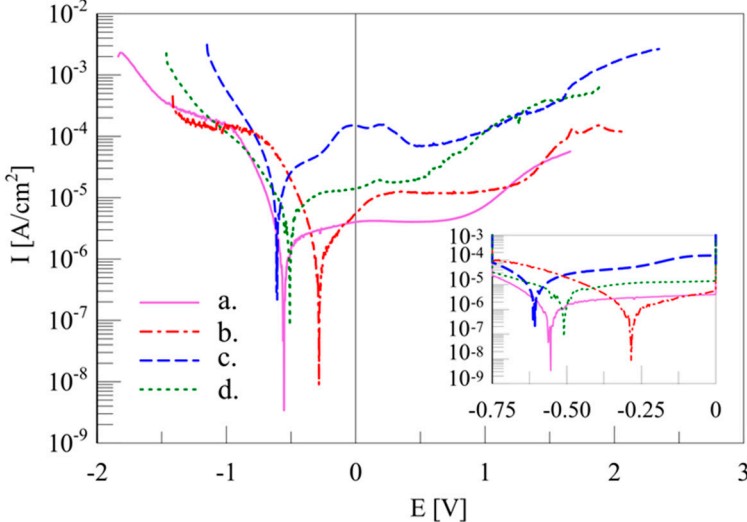

**Figure 5.** Potentiodynamic test result curves of Ti23Mo alloys obtained by different processing approaches in Ringer solution: Arc melted and annealed at 800 °C/24 h (**a**), MA for 48 h and hot pressed at 800 °C/5 min (**b**), cold-pressed and sintered at 800 °C/0.5 h (**c**) and cold-pressed with NH$_4$HCO$_3$ and sintered at 1150 °C/10 h (**d**).

Additionally, for the cold-pressed and sintered at 800 °C/0.5 h samples, surface functionalization by MAO and EPD was investigated. The structural analysis (Figure 6) of obtained modified surfaces confirms, beside strong Ti(β)substrate reflexes for MAO, a complex composition based on the oxides-Ti$_6$O (01-073-1118)/CaTiO$_3$ (01-075-0437), hydroxides-Ca(OH)$_2$ (04-014-7726), and apatite-Ca$_3$(PO$_4$)$_2$ (00-048-0488) mixture, as for EPD a fluorapatite layer (FA 01-071-0881). The morphological view of the samples revealed on SEM microphotographs (Figure 7), allows the characterization of the obtained surfaces as highly developed ones, with a specific formation. The EPD process fully obscures the MAO procedure, however, what remains important and confirmed in FA layer formation [51–53] is the influence of the substrate relation.

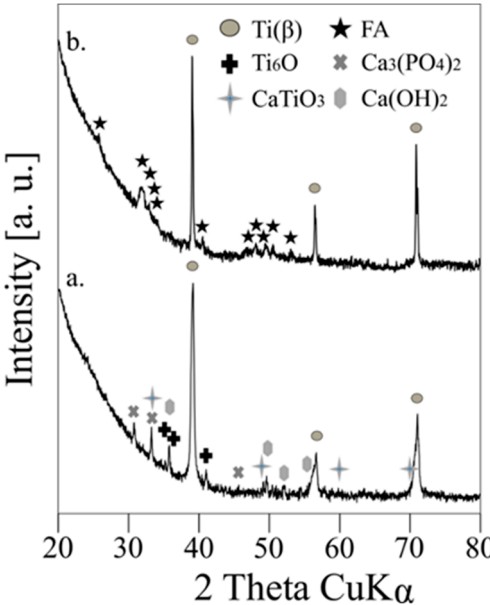

**Figure 6.** XRD spectra of Ti23Mo alloy MA for 48 h and cold-pressed and sintered at 800 °C/0.5 h and next surface-treated: MAO (**a**), MAO + EPD (**b**).

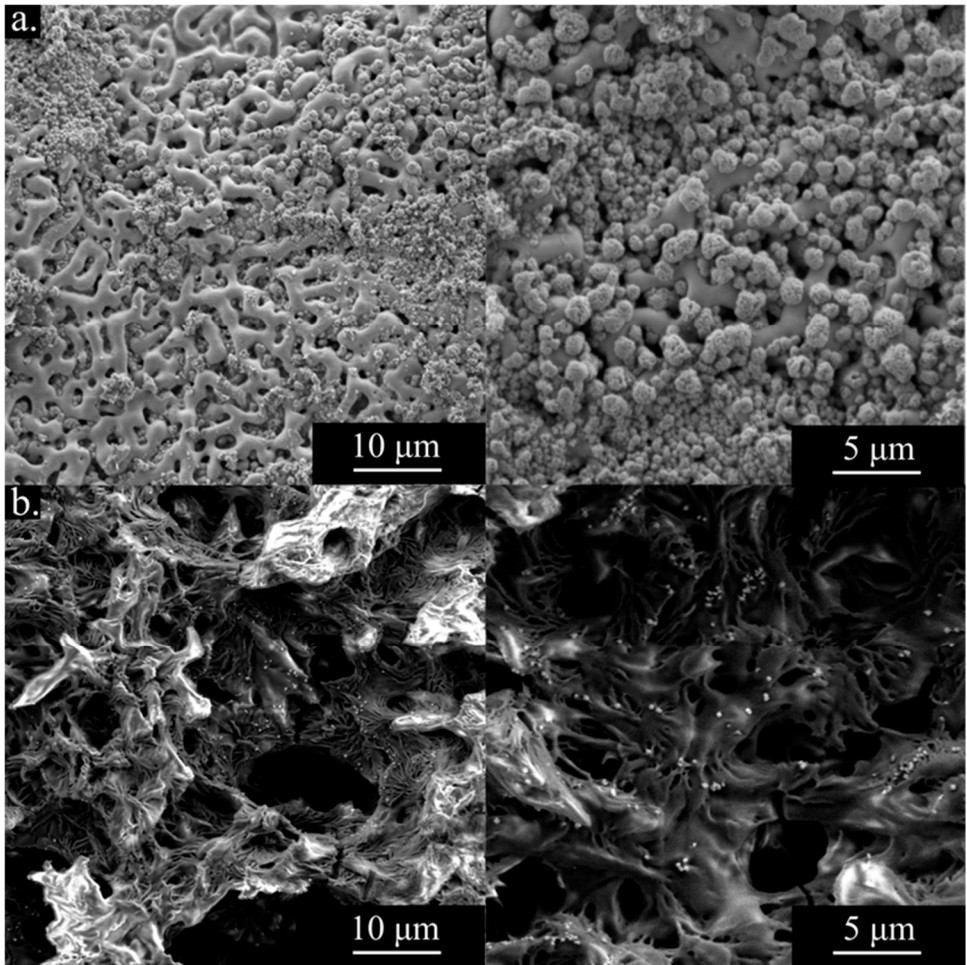

**Figure 7.** SEM micrographs of CP samples after surface treatment: MAO (**a**), MAO and EPD (**b**).

The evaluation of corrosion resistance in Ringer's solution (Table 5 and Figure 8) of modified layers shows visible improvement in accordance with the base sample. Also, what the collation of the potentiodynamic curves shows in Figure 8 is the specific sample behaviour in comparison to all analyzed processing examples.

**Table 5.** Estimated from Tafel extrapolation corrosion potential ($E_{corr}$) and current ($I_{corr}$) with calculated weight loss form Faraday law (W by polarization) and after 14-days immersing in Ringer solution environment (W by weight loss) of Ti23Mo alloys obtained by different processing approaches.

| Material | $E_{corr}$ (V) | $I_{corr}$ ($\mu A \cdot cm^{-2}$) | W by Polarization ($\mu g \cdot day^{-1}$) | W by Weight Loss ($\mu g \cdot day^{-1}$) |
|---|---|---|---|---|
| CP | −0.610(3) | 4.506(705) | 53.7(8) | 73.3(6) |
| CP + MAO | −0.194(2) | 0.8367(376) | 4.4(2) | 2.8(5) |
| CP + MAO + EPD | −0.615(6) | 7.519(725) | 39.5(4) | 6.9(3) |

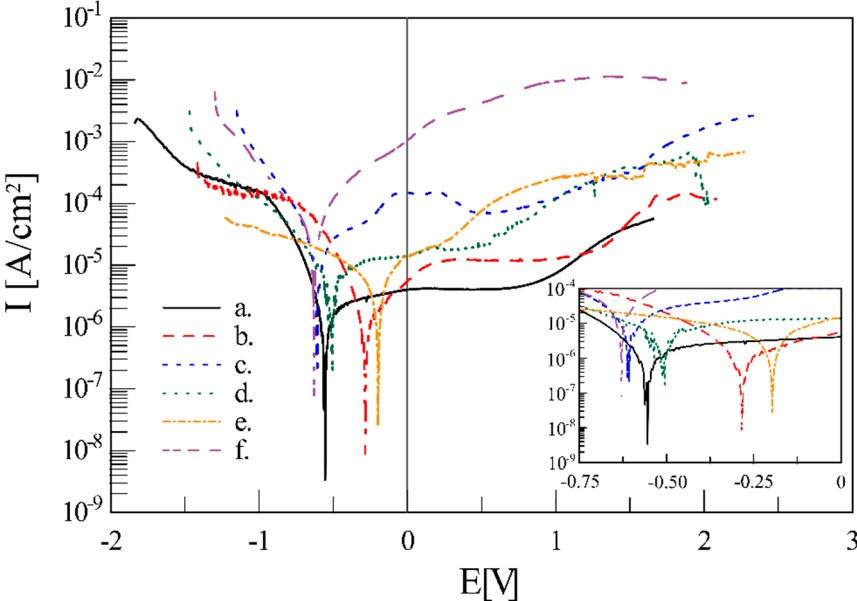

**Figure 8.** Potentiodynamic test result curves of Ti23Mo alloys obtained by different processing approaches in Ringer solution: Arc melted and annealed at 800 °C/24 h (**a**), MA for 48 h and hot pressed at 800 °C/5 min (**b**), cold-pressed and sintered at 800 °C/0.5 h (**c**), cold-pressed with NH$_4$HCO$_3$ and sintered at 1150 °C/10 h (**d**) and cold-pressed and sintered at 800 °C/0.5 h after surface treatment: MAO (**e**), MAO and EPD (**f**).

Finally, surface properties analysis based on the contact angle measurements in glycerol and diiodomethane testing fluids, and further estimation of the surface free energy with disperse and polar components, were performed on the prepared samples. Results (Table 6) confirmed that different processing approaches influence the SFE and it is dependent on structural and internal material characteristics. Secondly, for the additionally CP modified surfaces after MAO and MAO + EPD processes, we observed the decrease of SFE which for the proposed treatment justifies the investigated functionalization step. Low SFE corresponds to high wetting properties which for the hard tissue replacement application remain crucial, especially at the level of molecular activity at the interface region of the host.

**Table 6.** Contact angle (CA), surface free energy with disperse and polar components for Ti23Mo alloys obtained by different processing approaches.

| Material | Diiodomethane CA (°) | Glycerol CA (°) | Surface Free Energy (mN/m) | Disperse (mN/m) | Polar (mN/m) |
|---|---|---|---|---|---|
| AMA800 | 63.62 ± 9.44 | 62.72 ± 11.30 | 35.56 ± 6.31 | 27.84 ± 1.71 | 7.56 ± 5.26 |
| HP | 60.47 ± 5.68 | 69.98 ± 6.78 | 31.91 ± 3.16 | 28.14 ± 3.42 | 3.61 ± 1.94 |
| CP | 53.43 ± 13.61 | 28.09 ± 4.38 | 56.34 ± 1.81 | 32.30 ± 7.75 | 27.37 ± 4.00 |
| CP + NH$_4$HCO$_3$ | - | 54.34 ± 10.05 | - | - | - |
| CP + MAO | 64.64 ± 1.66 | 50.54 ± 14.46 | 42.56 ± 9.81 | 25.91 ± 0.94 | 16.65 ± 5.38 |
| CP + MAO + EPD | 54.93 ± 9.31 | 41.58 ± 3.35 | 48.84 ± 1.86 | 31.85 ± 5.31 | 16.99 ± 6.15 |

## 4. Conclusions

Conducted research allowed a synthesising of a new Ti23Mo alloy by the arc-melting, mechanical alloying, and powder metallurgy methods including cold and hot pressing approaches. Additionally, material volumetric and surface functionalization changes were also investigated. The influence of the processing approach on the phase transitions ($\alpha \rightarrow \beta$), microstructure, corrosion resistance, mechanical and surface wetting properties was studied. The following conclusions can be drawn:

(1)　-sintering of MA powder leads to the formation of the Ti($\beta$) based type alloys,

(2)　-the HP process at a low temperature (800 °C/5 min) of the Ti23Mo alloy in comparison to the cold pressing and sintering (800 °C/0.5 h) approach allows an obtainment of a low porosity high compactness pure Ti($\beta$) phase,

(3)　-the low-temperature sintering (below $\alpha \rightarrow \beta$ transus) allows the synthesizing of the bulk materials,

(4)　-the obtained microhardness test results favoured the samples with high compactness and low porosity,

(5)　-the indentation modulus and estimated sinters parameters obtained in this work confirm a relationship between the material phase and the internal structure,

(6)　-the potentiodynamic corrosion resistance analysis indicates a heavy dependence of the obtained results on the material's porosity and their chemical composition,

(7)　-the results obtained for surface modified MAO and MAO + EPD treatments confirms that the substrate has a crucial meaning for wetting and corrosion resistance characteristics

(8)　-the SFE, as the analysis confirms, stays strongly dependent on structural and internal material characteristics as dictated by different processing approaches.

**Author Contributions:** P.S., A.M., K.K. and M.J. conducted the experimental and analytical works as well as wrote the manuscript, M.J. supervised the project. All the authors contributed to the critical reading, and editing of the final version of the manuscript.

**Funding:** The research was financially supported by the Polish National Science Center DEC- 2017/25/B/ST8/02494).

**Conflicts of Interest:** The authors declare no conflict of interest.

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
