# Peer review of "Influence of the Processing Method on the Properties of Ti-23 at.% Mo Alloy"

_metals, doi:10.3390/met9090931_

Round 1

Reviewer 1 Report

- Both arc-melting (AM) and mechanical alloying (MA) used blended powders of Ti and Mo. What was the premix ratio?

- The virgin powders were about 44 μm in size, what were the powder size after mechanical alloying (MA)? This information is important for the readers to correlate the microstructure and property of hot-pressed (HP) and cold-pressed (CP) specimens.

- Specimens for CP + NH4HCO3 (cold pressed with NH4HCO3) were compacted at 400 MPa instead of 600 MPa as done for CP (cold pressed). Was this done intentionally to produce porous structure?

- Table 2 shows porosity determined from theoretical density (ρth) and calculated density (ρcal). What is the difference between theoretical and calculated density? How was the calculated density determined? Why didn’t you use actual measured density from the Archimedes method?

- Did you attempt to determine the green density of these samples before sintering?

- The manuscript lacks a clear flow of information and needs extensive English editing. 

Author Response

Rewiever 1

Comments and Suggestions for Authors

- Both arc-melting (AM) and mechanical alloying (MA) used blended powders of Ti and Mo. What was the premix ratio?

Response

The premix ratio corresponds to an atomic amount of elements for the base alloy proportion Mo -23% Ti-rest.

- The virgin powders were about 44 μm in size, what were the powder size after mechanical alloying (MA)? This information is important for the readers to correlate the microstructure and property of hot-pressed (HP) and cold-pressed (CP) specimens.

Response

The size of the powders after 48h of MA was 13.5 nm according to the Williamson-Hall approach calculation method and it was the subject of the detailed investigation with powder processing and preparation in our earlier research [18], the text was added see line 119.

- Specimens for CP + NH4HCO3 (cold pressed with NH4HCO3) were compacted at 400 MPa instead of 600 MPa as done for CP (cold pressed). Was this done intentionally to produce porous structure?

Response

The porous structures can be produced at much more lower pressure (400 MPa). To large pressure made some problems with compaction with NH4HCO3, so the authors decided to decrease it.

- Table 2 shows porosity determined from theoretical density (ρth) and calculated density (ρcal). What is the difference between theoretical and calculated density? How was the calculated density determined? Why didn’t you use actual measured density from the Archimedes method?

Response

Calculated density comes from the drainage method (so it was used) and the theoretical one (also from calculations) from the rule of the mixtures. The difference in estimation comes from the chosen method and for the porous samples, the mixed approach seems to be the best.

- Did you attempt to determine the green density of these samples before sintering?

Response

The ‘green density’ of all samples was no more than 50% of theoretical density. The authors didn’t feel that this information could bring a significant breakthrough.

- The manuscript lacks a clear flow of information and needs extensive English editing. 

Response

The manuscript was checked and corrected by a native English speaker.

The revised manuscript has been resubmitted to Metals.  We hope that our manuscript meets now Yours and the Reviewer's approval.

Sincerely

M. Jurczyk

Poznan University of Technology

Reviewer 2 Report

The article with the title "  Influence of the processing method on the properties 2 of Ti-23 at.% Mo alloy " is , unfortunately, to the limit of being publishable;  despite of the many experimental results presented it is missing a directory ideea; what intentions  exist? In the present form, in the paper there is a mixture of scientific results,  being impossible to identify  a clear scientific research  goal. The presentation  jumps steeply from an idea to a different one;

Plus ,a series of punctual arguments below, in no particular order:

- Why MA is not present in the list for  the methods of synthesis for  the Ti-23 at. Alloy  ? Using different methods, is the chemical composition similar?

 -In table 1, 2 and 3 the data for MA are missing.

- In fig. 3 , only the EDS spectrum of Ti23Mo hot pressed (800°C/5 min) is presented.

- The surface treatment functionalization based on MAO and EPD process could be a separate subject.

Author Response

Rewiever 2

Comments and Suggestions for Authors

The article with the title "  Influence of the processing method on the properties 2 of Ti-23 at.% Mo alloy " is , unfortunately, to the limit of being publishable;  despite of the many experimental results presented it is missing a directory ideea; what intentions  exist? In the present form, in the paper there is a mixture of scientific results,  being impossible to identify  a clear scientific research  goal. The presentation  jumps steeply from an idea to a different one;

Response:

Thank you for the comments.  

Recently, Ti-xx at % Mo (xx = 10 - 35) alloys has been prepared by mechanical alloying and powder metallurgy approach [18, 29]. Conducted research allows a synthesis of a new Ti23Mo alloy by the arc-melting, mechanical alloying and powder metallurgy methods including cold and hot pressing approach. Additionally, for the Ti23Mo alloy, the micro-arc oxidation (MAO) and electrophoretic deposition (EPD) approach were applied and led to the formation of apatite and fluorapatite (FA) layers, which improves analyzed surface properties compared to the base sample. The crystal structure, microstructure, composition, porosity, corrosion resistance, mechanical and surface wetting properties of the bulk synthesized alloy were studied.

(see also the graphical abstract, below). 

Plus ,a series of punctual arguments below, in no particular order:

- Why MA is not present in the list for  the methods of synthesis for  the Ti-23 at. Alloy  ? Using different methods, is the chemical composition similar?

Response

The MA and powder processing and preparation was the subject of detailed investigations in our previous study - see reference [18].

Additionally, new sentence was added see line 119

The size of the powders after 48h of MA was 13.5 nm according to the Williamson-Hall approach calculation method and it was the subject of the detailed investigation with powder processing and preparation in our earlier research [18].

Why MA is not present in the list for  the methods of synthesis for  the Ti-23 at. alloy  ? Using different methods, is the chemical composition similar?

The bulk or porous Ti-23 at. Mo alloy can’t be processed by MA. The application of MA can produce only powdered alloys, which was a starting powdered materials used for the synthesis of bulk or porous alloys. So, MA is not present in the list.

The chemical composition (Ti-23 at. Mo) is similar by the application of different methods – conformed by EDS studies, see new Fig. 3

 -In table 1, 2 and 3 the data for MA are missing.

Response

The MA was the subject of detailed investigation with powder processing and preparation in our earlier research [18].  – see line 119

- In fig. 3 , only the EDS spectrum of Ti23Mo hot pressed (800°C/5 min) is presented.

Response

Thank you. According to the reviewer suggestion, new Fig. 3 with all EDS spectra was added.

New Fig. 3 caption: 

Figure 3. The amount of elements in  Ti23Mo alloy: arc melted (a), hot pressed (800°C/5 min) (b), cold-pressed at 800°C/0.5 h (c), cold-pressed with NH4HCO3 at 1150°C/10 h (d) with their EDS spectra.

- The surface treatment functionalization based on MAO and EPD process could be a separate subject.

Response

The MAO and EPD process could be a separate subject however the author’s intention was to show a possible treatment approach and processing method influence the final material properties.

These processes led to the formation of apatite and fluorapatite layers. The results obtained for surface modified by MAO and MAO+EPD treatment confirms that the substrate has a crucial meaning for the wetting and the corrosion resistance characteristics.

The revised manuscript has been resubmitted to Metals.  We hope that our manuscript meets now Yours and the Reviewer's approval.

Sincerely

M. Jurczyk

Poznan University of Technology

Round 2

Reviewer 1 Report

Please specify the oxygen content of the powders and the purity of argon used in MA. These parameters have detrimental effects on mechanically alloyed powders, and can influence the final microstructure and mechanical properties. Line 17: for clarity, replace 'sintering' with 'inductively hot-pressed'. This is to avoid confusion with conventional sintering which was done for 30 min. Authors' revised manuscript is acknowledged, but there are still numerous grammatical/spelling errors. Just to mention a few Line 26, replace 'not modify' with 'unmodified' Line 62 and 63: Rephrase '... were also confirmed the highest values were obtained ...'? Line 79: replace '... alloys has been ... ' with '... alloys have been ...' Line 85: replace '... approach were applied ...' with  '... approaches were applied ...' Line 146: replace '... structure examination .... were ...' with '... structure examination .... was ...' Line 214 and 215: rephrase '... which for (110) plane reveals in the spectra.'? Line 420: replace 'bellow' with 'below' etc.

Author Response

Dear Reviewer,

We appreciate the opportunity to resubmit our manuscript entitled "Influence of the processing method on the properties of Ti-23 at.% Mo alloy”, to be considered for publication in Metals.

Thank you very much for your comments and suggestions that help considerably to improve the manuscript. We have made minor revision to the submitted article (changes are marked in red in the revised manuscript), with answers to the comments point by point below.

Responses and revision notes:

Rewiever 1

Comments and Suggestions for Authors

Please specify the oxygen content of the powders and the purity of argon used in MA. These parameters have detrimental effects on mechanically alloyed powders, and can influence the final microstructure and mechanical properties.

The MA was performed under Ar (99.999 % putity) by the application of the SPEX 8000 Mixer Mill. The total milling time was 48 h. The Ti and Mo powders were weighed, blended and insert into stainless steel vials in the glove box (LabMaster 130) filled with automatically controlled argon atmosphere (O2 < 2 ppm and H2O < 1 ppm). .

We agree with the Referee's comments - the  oxygen content in final produced samples can  influence their final microstructure and mechanical properties.

See Introduction – pp 60 - 72

Line 17: for clarity, replace 'sintering' with 'inductively hot-pressed'. This is to avoid confusion with conventional sintering which was done for 30 min.

Thank you, the wortd “sintering” was replaced

Authors' revised manuscript is acknowledged, but there are still numerous grammatical/spelling errors. Just to mention a few

Line 26, replace 'not modify' with 'unmodified'

Thank you, the wortd “sintering” was replaced

Line 62 and 63: Rephrase '... were also confirmed the highest values were obtained ...'?

Thank you, some part of the sentence was deleted.

A strong influence of the structure on elasticity modulus and hardness were also confirmed the highest values were obtained for aged at 450°C samples, following the α and ω phase composition.

Line 79: replace '... alloys has been ... ' with '... alloys have been ...'

Thank you – alloys have been

Line 85: replace '... approach were applied ...' with  '... approaches were applied ...'

Thank you - approaches were applied

Line 146: replace '... structure examination .... were ...' with '... structure examination .... was ...'

Thank you – structure examination .... was   -  (line 149)

Line 214 and 215: rephrase '... which for (110) plane reveals in the spectra.'?

Thank you – some part of the sentence was removed

After 5 h of milling the (110) plane from the regular structure of molybdenum disappears and a new MoTi phase is formed.

Line 420: replace 'bellow' with 'below' etc.

Thank you – below   -  (line 424)

The revised manuscript has been resubmitted to Metals.  We hope that our manuscript meets now Yours and the Reviewer's approval.

Sincerely

M. Jurczyk

Poznan University of Technology
